# Nearly tight sample complexity bounds for learning mixtures of Gaussians via sample compression schemes[*]

**Hassan Ashtiani**
Department of Computing and Software
McMaster University, and
Vector Institute, ON, Canada
`zokaeiam@mcmaster.ca`

**Shai Ben-David**
School of Computer Science
University of Waterloo,
Waterloo, ON, Canada
`shai@uwaterloo.ca`

**Nicholas J. A. Harvey**
Department of Computer Science
University of British Columbia
Vancouver, BC, Canada
`nickhar@cs.ubc.ca`

**Christopher Liaw**
Department of Computer Science
University of British Columbia
Vancouver, BC, Canada
`cvliaw@cs.ubc.ca`

**Abbas Mehrabian**
School of Computer Science
McGill University
Montréal, QC, Canada
`abbasmehrabian@gmail.com`

**Yaniv Plan**
Department of Mathematics
University of British Columbia
Vancouver, BC, Canada
`yaniv@math.ubc.ca`

## Abstract

We prove that $\widetilde{\Theta}(kd^2/\varepsilon^2)$ samples are necessary and sufficient for learning a mixture of $k$ Gaussians in $\mathbb{R}^d$, up to error $\varepsilon$ in total variation distance. This improves both the known upper bounds and lower bounds for this problem. For mixtures of axis-aligned Gaussians, we show that $\widetilde{O}(kd/\varepsilon^2)$ samples suffice, matching a known lower bound.

The upper bound is based on a novel technique for distribution learning based on a notion of *sample compression*. Any class of distributions that allows such a sample compression scheme can also be learned with few samples. Moreover, if a class of distributions has such a compression scheme, then so do the classes of *products* and *mixtures* of those distributions. The core of our main result is showing that the class of Gaussians in $\mathbb{R}^d$ has a small-sized sample compression.

## 1 Introduction

Estimating distributions from observed data is a fundamental task in statistics that has been studied for over a century. This task frequently arises in applied machine learning and it is common to assume that the distribution can be modeled using a mixture of Gaussians. Popular software packages have implemented heuristics, such as the EM algorithm, for learning a mixture of Gaussians. The theoretical machine learning community also has a rich literature on distribution learning; the recent survey [9] considers learning structured distributions, and the survey [13] focuses on mixtures of Gaussians.

This paper develops a general technique for distribution learning, then employs this technique in the important setting of mixtures of Gaussians. The theoretical model we adopt is *density estimation*:

---

[*]For the full version of this paper see [2].

given i.i.d. samples from an unknown target distribution, find a distribution that is close to the target distribution in *total variation (TV) distance*. Our focus is on sample complexity bounds: using as few samples as possible to obtain a good estimate of the target distribution. For background on this model see, e.g., [7, Chapter 5] and [9].

Our new technique for proving upper bounds on the sample complexity involves a form of *sample compression*. If it is possible to "encode" members of a class of distributions using a carefully chosen subset of the samples, then this yields an upper bound on the sample complexity of distribution learning for that class. In particular, by constructing compression schemes for mixtures of axis-aligned Gaussians and general Gaussians, we obtain new upper bounds on the sample complexity of learning with respect to these classes, which we prove to be optimal up to logarithmic factors.

## 1.1 Main results

In this section, all learning results refer to the problem of producing a distribution within total variation distance $\varepsilon$ from the target distribution. Our first main result is an upper bound for learning mixtures of multivariate Gaussians. This bound is tight up to logarithmic factors.

**Theorem 1.1** *The class of $k$-mixtures of $d$-dimensional Gaussians can be learned using $\widetilde{O}(kd^2/\varepsilon^2)$ samples.*

We emphasize that the $\widetilde{O}(\cdot)$ notation hides a factor $\text{polylog}(kd/\varepsilon)$, but has *no dependence whatsoever on the condition number* or scaling of the distribution. Previously, the best known upper bounds on the sample complexity of this problem were $\widetilde{O}(kd^2/\varepsilon^4)$, due to [3], and $O(k^4d^4/\varepsilon^2)$, based on a VC-dimension bound that we discuss below. For the case of a single Gaussian (i.e., $k = 1$), a sample complexity bound of $O(d^2/\varepsilon^2)$ is well known, again using a VC-dimension bound discussed below.

Our second main result is a minimax lower bound matching Theorem 1.1 up to logarithmic factors.

**Theorem 1.2** *Any method for learning the class of $k$-mixtures of $d$-dimensional Gaussians has sample complexity $\Omega(kd^2/\varepsilon^2 \log^3(1/\varepsilon)) = \widetilde{\Omega}(kd^2/\varepsilon^2)$.*

Here and below $\widetilde{\Omega}$ (and $\widetilde{O}$) allow for poly-logarithmic factors. Previously, the best known lower bound on the sample complexity was $\widetilde{\Omega}(kd/\varepsilon^2)$ [20]. Even for a single Gaussian (i.e., $k = 1$), an $\widetilde{\Omega}(d^2/\varepsilon^2)$ lower bound was not known prior to this work.

Our third main result is an upper bound for learning mixtures of *axis-aligned* Gaussians, i.e., Gaussians with diagonal covariance matrices. This bound is tight up to logarithmic factors.

**Theorem 1.3** *The class of $k$-mixtures of axis-aligned $d$-dimensional Gaussians can be learned using $\widetilde{O}(kd/\varepsilon^2)$ samples.*

A matching lower bound of $\widetilde{\Omega}(kd/\varepsilon^2)$ was proved in [20]. Previously, the best known upper bounds were $\widetilde{O}(kd/\varepsilon^4)$, due to [3], and $O((k^4d^2 + k^3d^3)/\varepsilon^2)$, based on a VC-dimension bound that we discuss below.

**Computational efficiency.** Although our approach for proving sample complexity upper bounds is algorithmic, our focus is not on computational efficiency. The resulting algorithms have nearly optimal sample complexities, but their running times are exponential in the dimension $d$ and the number of mixture components $k$. More precisely, the running time is $2^{kd^2 \, \text{polylog}(d,k,1/\varepsilon)}$ for mixtures of general Gaussians, and $2^{kd \, \text{polylog}(d,k,1/\varepsilon)}$ for mixtures of axis-aligned Gaussians. The existence of a polynomial time algorithm for density estimation is unknown even for the class of mixtures of axis-aligned Gaussians, see [10, Question 1.1].

Even for the case of a single Gaussian, the published proofs of the $O(d^2/\varepsilon^2)$ bound (of which we are aware) are not algorithmically efficient. Using ideas from our proof of Theorem 1.1, in the full version we show that an algorithmically efficient proof for single Gaussians can be obtained by computing the empirical mean and a careful modification of the sample covariance matrix of $O(d^2/\varepsilon^2)$ samples.

## 1.2 Related work

Distribution learning is a vast topic and many approaches have been considered in the literature; here we only review approaches that are most relevant to our problem.

For parametric families of distributions, a common approach is to use the samples to estimate the parameters of the distribution, possibly in a maximum likelihood sense, or possibly aiming to approximate the true parameters. For the specific case of mixtures of Gaussians, there is a substantial theoretical literature on algorithms that approximate the mixing weights, means and covariances; see [13] for a recent survey of this literature. The strictness of this objective cuts both ways. On the one hand, a successful learner uncovers substantial structure of the target distribution. On the other hand, this objective is clearly impossible when the means and covariances are extremely close. Thus, algorithms for parameter estimation of mixtures necessarily require some "separability" assumptions on the target parameters.

Density estimation has a long history in the statistics literature, where the focus is on the sample complexity question; see [6, 7, 19] for general background. It was first studied in the computational learning theory community under the name *PAC learning of distributions* by [14], whose focus is on the computational complexity of the learning problem.

For density estimation there are various possible measures of distance between distributions, the most popular ones being the TV distance and the Kullback-Leibler (KL) divergence. Here we focus on the TV distance since it has several appealing properties, such as being a metric and having a natural probabilistic interpretation. In contrast, KL divergence is not even symmetric and can be unbounded even for intuitively close distributions. For a detailed discussion on why TV is a natural choice, see [7, Chapter 5].

A popular method for distribution learning in practice is kernel density estimation (see, e.g., [7, Chapter 9]). The few rigorously proven sample complexity bounds for this method require either smoothness assumptions (e.g., [7, Theorem 9.5]) or boundedness assumptions (e.g., [12, Theorem 2.2]) on the class of densities. The class of Gaussians is not universally Lipschitz or universally bounded, so those results do not apply to the problems we consider.

Another elementary method for density estimation is using histogram estimators (see, e.g., [7, Section 10.3]). Straightforward calculations show that histogram estimators for mixtures of Gaussians result in a sample complexity that is exponential in the dimension. The same is true for estimators based on piecewise polynomials.

The *minimum distance estimate* [7, Section 6.8] is another approach for deriving sample complexity upper bounds for distribution learning. This approach is based on uniform convergence theory. In particular, an upper bound for any class of distributions can be achieved by bounding the VC-dimension of an associated set system, called the Yatracos class (see [7, page 58] for the definition). For example, [11] used this approach to bound the sample complexity of learning high-dimensional log-concave distributions. For the class of single Gaussians in $d$ dimensions, this approach leads to the optimal sample complexity upper bound of $O(d^2/\varepsilon^2)$. However, for mixtures of Gaussians and axis-aligned Gaussians in $\mathbb{R}^d$, the best known VC-dimension bounds [1, Theorem 8.14] result in loose upper bounds of $O(k^4 d^4/\varepsilon^2)$ and $O((k^4 d^2 + k^3 d^3)/\varepsilon^2)$, respectively.

Another approach is to first approximate the mixture class using a more manageable class such as piecewise polynomials, and then study the associated Yatracos class, see, e.g., [5]. However, piecewise polynomials do a poor job in approximating $d$-dimensional Gaussians, resulting in an exponential dependence on $d$.

For density estimation of mixtures of Gaussians, the current best sample complexity upper bounds (in terms of $k$ and $d$) are $\widetilde{O}(kd^2/\varepsilon^4)$ for general Gaussians and $\widetilde{O}(kd/\varepsilon^4)$ for axis-aligned Gaussians, both due to [3]. For the general Gaussian case, their method takes an i.i.d. sample of size $\widetilde{O}(kd^2/\varepsilon^2)$ and partitions this sample in every possible way into $k$ subsets. Based on those partitions, $k^{\widetilde{O}(kd^2/\varepsilon^2)}$ "candidate distributions" are generated. The problem is then reduced to learning with respect to this finite class of candidates. Their sample complexity has a suboptimal factor of $1/\varepsilon^4$, of which $1/\varepsilon^2$ arises in their approach for choosing the best candidate, and another factor $1/\varepsilon^2$ is due to the exponent in the number of candidates.

Our approach via compression schemes also ultimately reduces the problem to learning with respect to finite classes. However, our compression technique leads to a more refined bound. In the case of mixtures of Gaussians, one factor of $1/\varepsilon^2$ is again incurred due to learning with respect to finite classes. The key is that the number of compressed samples has *no* additional factor of $1/\varepsilon^2$, so the overall sample complexity bound has only an $\widetilde{O}(1/\varepsilon^2)$ dependence on $\varepsilon$.

As for lower bounds on the sample complexity, much fewer results are known for learning mixtures of Gaussians. The only lower bound of which we are aware is due to [20], which shows a bound of $\widetilde{\Omega}(kd/\varepsilon^2)$ for learning mixtures of spherical Gaussians (and hence for general Gaussians as well). This bound is tight for the axis-aligned case, as we show in Theorem 1.3, but loose in the general case, as we show in Theorem 1.2.

### 1.2.1 Comparison to parameter estimation

In this section we observe that neither our upper bound (Theorem 1.1) nor our lower bound (Theorem 1.2) can directly follow from bounds on parameter estimation for Gaussian distributions. Recall that our sample complexity upper bound in Theorem 1.1 has no dependence on the condition number of the distribution. We now show that, if a learning algorithm with entrywise approximation guarantee is used to learn the distribution in KL divergence or TV distance, then the approximation parameter must depend on the condition number. Let $\kappa(\Sigma)$ be the condition number of the covariance matrix $\Sigma$, i.e., the ratio of the maximum and minimum eigenvalues; refer to Section 2 for other relevant definitions.

**Proposition 1.4** *Set $\varepsilon = \frac{2}{\kappa(\Sigma)+1}$. There exist two covariance matrices $\Sigma$ and $\hat{\Sigma}$ that are good entrywise approximations:*

$$|\Sigma_{i,j} - \hat{\Sigma}_{i,j}| \le \varepsilon \qquad and \qquad \hat{\Sigma}_{i,j} \in [1, 1+2\varepsilon] \cdot \Sigma_{i,j} \qquad \forall i,j,$$

*but the corresponding distributions are as far as they can get, i.e.,*

$$\mathrm{KL}\left(\mathcal{N}(0,\Sigma) \,\|\, \mathcal{N}(0,\hat{\Sigma})\right) = \infty \qquad and \qquad \mathrm{TV}\left(\mathcal{N}(0,\Sigma), \mathcal{N}(0,\hat{\Sigma})\right) = 1.$$

Thus, given a black-box algorithm that provides an entrywise $\varepsilon$-approximation to the true covariance matrix $\Sigma$, if $\varepsilon \ge \frac{2}{\kappa(\Sigma)+1}$, it might output $\hat{\Sigma}$, which does not approximate $\Sigma$ in KL divergence or total variation distance.

One might imagine that lower bounds on the sample complexity of parameter estimation readily imply lower bounds on distribution learning. The following proposition shows this is not the case.

**Proposition 1.5** *For any $\varepsilon \in (0, 1/2]$ there exist two covariance matrices $\Sigma$ and $\hat{\Sigma}$ such that $\mathrm{TV}\left(\mathcal{N}(0,\Sigma), \mathcal{N}(0,\hat{\Sigma})\right) \le \varepsilon$, but there exist $i,j$ such that, for any $c \ge 1$, $\hat{\Sigma}_{i,j} \notin [1/c, c] \cdot \Sigma_{i,j}$.*

### 1.3 Our techniques

We introduce a method for learning distributions via a novel form of *compression*. Given a class of distributions, suppose there is a method for "compressing" information about the true distribution using a mix of samples from that distribution and some additional bits. Further, suppose there exists a fixed (deterministic) *decoder* for the class, such that given the samples and additional bits, it approximately recovers the original distribution. In this case, if the size of the compressed set and the number of bits is guaranteed to be small, we show that the sample complexity of learning that class is small as well.

More precisely, we say a class of distributions *admits $(\tau, t, m)$ compression* if there exists a *decoder function* such that upon generating $m$ i.i.d. samples from any distribution in the class, we are guaranteed, with reasonable probability, to have a subset of size at most $\tau$ of that sample, and a sequence of at most $t$ bits, on which the decoder outputs an approximation to the original distribution. Note that $\tau$, $t$, and $m$ can be functions of $\varepsilon$, the accuracy parameter.

We prove that compression implies learning. In particular, if a class admits $(\tau, t, m)$ compression, then the sample complexity of learning with respect to this class is bounded by $\widetilde{O}(m + (\tau + t)/\varepsilon^2)$ (Theorem 3.5).

An attractive property of compression is that it enjoys two closure properties. Specifically, if a base class admits compression, then the class of mixtures of that base class, as well as the class of products of the base class, are compressible (Lemmas 3.6 and 3.7).

Consequently, it suffices to provide a compression scheme for the class of single Gaussian distributions in order to obtain a compression scheme for the class of mixtures of Gaussians (and therefore, to be able to bound their sample complexity). We prove that the class of $d$-dimensional Gaussian distributions admits $(\widetilde{O}(d), \widetilde{O}(d^2), \widetilde{O}(d))$ compression (Lemma 4.1). The high level idea is that by generating $\widetilde{O}(d)$ samples from a Gaussian, one can get a rough sketch of the geometry of the Gaussian. In particular, the points drawn from a Gaussian concentrate around an ellipsoid centered at the mean and whose principal axes are the eigenvectors of the covariance matrix. Using ideas from convex geometry and random matrix theory, we show one can in fact encode the center of the ellipsoid *and* the principal axes using a linear combination of these samples. Then we discretize the coefficients and obtain an approximate encoding.

The above results together imply tight (up to logarithmic factors) upper bounds of $\widetilde{O}(kd^2/\varepsilon^2)$ for mixtures of $k$ Gaussians, and $\widetilde{O}(kd/\varepsilon^2)$ for mixtures of $k$ axis-aligned Gaussians over $\mathbb{R}^d$. The *compression framework* we introduce is quite flexible, and can be used to prove sample complexity upper bounds for other distribution classes as well. This is left for future work.

In this paper we assume the target belongs to some known class of distributions (this is called the realizable setting in the learning theory literature). In the full version of this paper [2] we relax this requirement and give similar sample complexity bounds for the setting where the target is close (in TV distance) to some distribution in the class (known as agnostic learning).

**Lower bound.**    For proving our lower bound for mixtures of Gaussians, we first prove a lower bound of $\widetilde{\Omega}(d^2/\varepsilon^2)$ for learning a single Gaussian. Although the approach is quite intuitive, the details are intricate and much care is required to make a formal proof. The main step is to construct a large family (of size $2^{\Omega(d^2)}$) of covariance matrices such that the associated Gaussian distributions are well-separated in terms of their TV distance while simultaneously ensuring that their relative KL divergences are small. Once this is established, we can then apply a generalized version of Fano's inequality to complete the proof.

To construct this family of covariance matrices, we sample $2^{\Omega(d^2)}$ matrices from the following probabilistic process: start with an identity covariance matrix; then choose a random subspace of dimension $d/9$ and slightly increase the eigenvalues corresponding to this eigenspace. It is easy to bound the KL divergences between the constructed Gaussians. To lower bound the total variation, we show that for every pair of these distributions, there is some subspace for which a vector drawn from one Gaussian will have slightly larger projection than a vector drawn from the other Gaussian. Quantifying this gap will then give us the desired lower bound on the total variation distance.

**Paper outline.**    We set up our formal framework and notation in Section 2. In Section 3, we define compression schemes for distributions, prove their closure properties, and show their connection with density estimation. Theorem 1.1 and Theorem 1.3 are proved in Section 4. The proof of Theorem 1.2 as well as all other omitted proofs can be found in the full version [2].

## 2   Preliminaries

A *distribution learning method* or *density estimation method* is an algorithm that takes as input a sequence of i.i.d. samples generated from a distribution $g$, and outputs (a description of) a distribution $\hat{g}$ as an estimation for $g$. We work with continuous distributions in this paper, and so we identify a probability distribution by its probability density function. Let $f_1$ and $f_2$ be two probability distributions defined over $\mathbb{R}^d$ and let $\mathcal{L}$ be the set of Lebesgue measurable subsets of $\mathbb{R}^d$. Their *total variation (TV) distance* is defined by

$$\mathrm{TV}\,(f_1, f_2) \;:=\; \sup_{B \in \mathcal{L}} \int_B \big(f_1(x) - f_2(x)\big)\mathrm{d}x \;=\; \frac{1}{2}\|f_1 - f_2\|_1\,,$$

where $\|f\|_1 := \int_{\mathbb{R}^d} |f(x)| \mathrm{d}x$ is the $L_1$ norm of $f$. The *Kullback-Leibler (KL) divergence* between $f_1$ and $f_2$ is defined by

$$\mathrm{KL}\left(f_1 \parallel f_2\right) := \int_{\mathbb{R}^d} f_1(x) \log \frac{f_1(x)}{f_2(x)} \mathrm{d}x.$$

In the following definitions, $\mathcal{F}$ is a class of probability distributions, and $g$ is a distribution (not necessarily in $\mathcal{F}$).

**Definition 2.1 ($\varepsilon$-approximation)** *A distribution $\hat{g}$ is an $\varepsilon$-approximation for $g$ if $\|\hat{g} - g\|_1 \leq \varepsilon$.*

**Definition 2.2 (PAC-learning distributions)** *A distribution learning method is called a (realizable) PAC-learner for $\mathcal{F}$ with sample complexity $m_{\mathcal{F}}(\varepsilon, \delta)$ if, for all distributions $g \in \mathcal{F}$ and all $\varepsilon, \delta \in (0, 1)$, given $\varepsilon$, $\delta$, and an i.i.d. sample of size $m_{\mathcal{F}}(\varepsilon, \delta)$ from $g$, with probability at least $1 - \delta$ (over the samples) the method outputs an $\varepsilon$-approximation of $g$.*

Let $\Delta_n := \{ (w_1, \ldots, w_n) : w_i \geq 0, \sum w_i = 1 \}$ denote the $n$-dimensional simplex.

**Definition 2.3 ($k$-mix($\mathcal{F}$))** *Let $\mathcal{F}$ be a class of probability distributions. Then the class of $k$-mixtures of $\mathcal{F}$, written $k$-mix($\mathcal{F}$), is defined as*

$$k\text{-mix}(\mathcal{F}) := \left\{ \sum_{i=1}^k w_i f_i : (w_1, \ldots, w_k) \in \Delta_k, f_1, \ldots, f_k \in \mathcal{F} \right\}.$$

Let $d$ denote the dimension. A Gaussian distribution with mean $\mu \in \mathbb{R}^d$ and covariance matrix $\Sigma \in \mathbb{R}^{d \times d}$ is denoted by $\mathcal{N}(\mu, \Sigma)$. If $\Sigma$ is a diagonal matrix, then $\mathcal{N}(\mu, \Sigma)$ is called an *axis-aligned* Gaussian. For a distribution $g$, we write $X \sim g$ to mean $X$ is a random variable with distribution $g$, and we write $S \sim g^m$ to mean that $S$ is an i.i.d. sample of size $m$ generated from $g$.

We will use $\|v\|$ or $\|v\|_2$ to denote the Euclidean norm of a vector $v$, $\|A\|$ or $\|A\|_2$ to denote the operator norm of a matrix $A$, and $\|A\|_F := \sqrt{\mathrm{Tr}(A^\mathsf{T} A)}$ to denote the Frobenius norm of a matrix $A$. For $x \in \mathbb{R}$, we will write $(x)_+ := \max\{0, x\}$. All logarithms are in the natural base.

## 3 Compression schemes and their connection with learning

Let $\mathcal{F}$ be a class of distributions over a domain $Z$.

**Definition 3.1 (distribution decoder)** *A distribution decoder for $\mathcal{F}$ is a deterministic function $\mathcal{J} : \bigcup_{n=0}^\infty Z^n \times \bigcup_{n=0}^\infty \{0, 1\}^n \to \mathcal{F}$, which takes a finite sequence of elements of $Z$ and a finite sequence of bits, and outputs a member of $\mathcal{F}$.*

**Definition 3.2 (distribution compression schemes)** *Let $\tau, t, m : (0, 1) \to \mathbb{Z}_{\geq 0}$ be functions. We say $\mathcal{F}$ admits $(\tau, t, m)$ compression if there exists a decoder $\mathcal{J}$ for $\mathcal{F}$ such that for any distribution $g \in \mathcal{F}$, the following holds:*

*For any $\varepsilon \in (0, 1)$, if a sample $S$ is drawn from $g^{m(\varepsilon)}$, then with probability at least $2/3$, there exists a sequence $L$ of at most $\tau(\varepsilon)$ elements of $S$, and a sequence $B$ of at most $t(\varepsilon)$ bits, such that $\|\mathcal{J}(L, B) - g\|_1 \leq \varepsilon$.*

Note that $S$ and $L$ are sequences rather than sets; in particular, they can contain repetitions. Also note that in this definition, $m(\varepsilon)$ is a *lower bound* on the number of samples needed, whereas $\tau(\varepsilon), t(\varepsilon)$ are *upper bounds* on the size of compression and the number of bits.

Essentially, the definition asserts that with reasonable probability, there is a (short) sequence consisting of elements $S$ and some (small number of) additional bits, from which $g$ can be approximately reconstructed. We say that the distribution $g$ is "encoded" with $L$ and $B$, and in general we would like to have a compression scheme of a small size.

**Remark 3.3** *In the definition above we required the probability of existence of $L$ and $B$ to be at least $2/3$, but one can boost this probability to $1 - \delta$ by generating a sample of size $m(\varepsilon) \log(1/\delta)$.*

Next we show that if a class of distributions can be compressed, then it can be learned; thus we build the connection between compression and learning. We will need the following useful result about

PAC-learning of finite classes of distributions, which immediately follows from [7, Theorem 6.3] and a standard Chernoff bound. It states that a finite class of size $M$ can be learned using $O(\log(M/\delta)/\varepsilon^2)$ samples. Denote by $[M]$ the set $\{1, 2, ..., M\}$. Throughout the paper, $a/bc$ always means $a/(bc)$.

**Theorem 3.4 ([7])** *There exists a deterministic algorithm that, given candidate distributions $f_1, \ldots, f_M$, a parameter $\varepsilon > 0$, and $\log(3M^2/\delta)/2\varepsilon^2$ i.i.d. samples from an unknown distribution $g$, outputs an index $j \in [M]$ such that*

$$\|f_j - g\|_1 \le 3 \min_{i \in [M]} \|f_i - g\|_1 + 4\varepsilon,$$

*with probability at least $1 - \delta/3$.*

The proof of the following theorem appears in the full version [2].

**Theorem 3.5 (compressibility implies learnability)** *Suppose $\mathcal{F}$ admits $(\tau, t, m)$ compression. Let $\tau'(\varepsilon) := \tau(\varepsilon/6) + t(\varepsilon/6)$. Then $\mathcal{F}$ can be learned using*

$$O\left(m\left(\frac{\varepsilon}{6}\right)\log\left(\frac{1}{\delta}\right) + \frac{\tau'(\varepsilon)\log(m(\frac{\varepsilon}{6})\log(1/\delta)) + \log(1/\delta)}{\varepsilon^2}\right) = \widetilde{O}\left(m\left(\frac{\varepsilon}{6}\right) + \frac{\tau'(\varepsilon)}{\varepsilon^2}\right) \text{ samples.}$$

We next prove two closure properties of compression schemes. First, Lemma 3.6 below states that if a class $\mathcal{F}$ of distributions can be compressed, then the class of distributions that are formed by taking products of members of $\mathcal{F}$ can also be compressed. If $p_1, \ldots, p_d$ are distributions over domains $Z_1, \ldots, Z_d$, then $\prod_{i=1}^d p_i$ denotes the standard product distribution over $\prod_{i=1}^d Z_i$. For a class $\mathcal{F}$ of distributions, define $\mathcal{F}^d := \left\{\prod_{i=1}^d p_i : p_1, \ldots, p_d \in \mathcal{F}\right\}$.

**Lemma 3.6 (compressing product distributions)** *If $\mathcal{F}$ admits $(\tau(\varepsilon), t(\varepsilon), m(\varepsilon))$ compression, then $\mathcal{F}^d$ admits $(d\tau(\varepsilon/d), dt(\varepsilon/d), m(\varepsilon/d)\log(3d))$ compression.*

Our next lemma states that if a class $\mathcal{F}$ of distributions can be compressed, then the class of distributions that are formed by taking mixtures of members of $\mathcal{F}$ can also be compressed.

**Lemma 3.7 (compressing mixtures)** *If $\mathcal{F}$ admits $(\tau(\varepsilon), t(\varepsilon), m(\varepsilon))$ compression, then $k$-mix$(\mathcal{F})$ admits $(k\tau(\varepsilon/3), kt(\varepsilon/3) + k\log_2(4k/\varepsilon)), 48m(\varepsilon/3)k\log(6k)/\varepsilon)$ compression.*

# 4 Upper bound: learning mixtures of Gaussians by compression schemes

In this section we prove an upper bound of $\widetilde{O}(kd^2/\varepsilon^2)$ for the sample complexity of learning mixtures of $k$ Gaussians in $d$ dimensions, and an upper bound of $\widetilde{O}(kd/\varepsilon^2)$ for the sample complexity of learning mixtures of $k$ axis-aligned Gaussians. The heart of the proof is to show that Gaussians have compression schemes in any dimension.

**Lemma 4.1** *For any positive integer $d$, the class of $d$-dimensional Gaussians admits an $\left(O(d\log(2d)), O(d^2\log(2d)\log(d/\varepsilon)), O(d\log(2d))\right)$ compression scheme.*

**Remark 4.2** *In the special case $d = 1$, there also exists a $(2, 0, O(1/\varepsilon))$ (i.e. constant size) compression scheme: if we draw $C/\varepsilon$ samples from $\mathcal{N}(\mu, \sigma^2)$, for a sufficiently large constant $C$, with probability at least $2/3$ there exist two points in the sample such that one of them is within distance $\sigma\varepsilon/2$ of $\mu - \sigma$ and the other one is within distance $\sigma\varepsilon/2$ of $\mu + \sigma$. Given these two points, the decoder can estimate $\mu$ and $\sigma$ up to additive precision $\varepsilon\sigma/2$, which results in an $\varepsilon$-approximation of $\mathcal{N}(\mu, \sigma^2)$ in total variation distance. Remarkably, this compression scheme has constant size, as the value of $\tau + t$ is independent of $\varepsilon$ (unlike Lemma 4.1). This scheme can be used instead of Lemma 4.1 in the proof of Theorem 1.3, although it would not improve the sample complexity bound asymptotically.*

**Proof of Theorem 1.1.** Combining Lemma 4.1 and Lemma 3.7 implies that the class of $k$-mixtures of $d$-dimensional Gaussians admits an

$$\left(O(kd\log(2d)), \, O(kd^2\log(2d)\log(d/\varepsilon) + k\log(k/\varepsilon)), \, O(dk\log k\log(2d)/\varepsilon)\right)$$

compression scheme. Applying Theorem 3.5 with $m(\varepsilon) = \widetilde{O}(dk/\varepsilon)$ and $\tau'(\varepsilon) = \widetilde{O}(d^2k)$ shows that the sample complexity of learning this class is $\widetilde{O}(kd^2/\varepsilon^2)$. This proves Theorem 1.1. ∎

**Proof of Theorem 1.3.** Let $\mathcal{G}$ denote the class of 1-dimensional Gaussian distributions. By Lemma 4.1, $\mathcal{G}$ admits an $(O(1), O(\log(1/\varepsilon)), O(1))$ compression scheme. Combining Lemma 3.6 and Lemma 3.7 gives the class $k\text{-mix}(\mathcal{G}^d)$ admits $(O(kd), O(kd\log(d/\varepsilon) + k\log(k/\varepsilon)), O(k\log(k)\log(3d)/\varepsilon))$ compression. Applying Theorem 3.5 implies that the class of $k$-mixtures of axis-aligned Gaussians in $\mathbb{R}^d$ can be learned using $\widetilde{O}(kd/\varepsilon^2)$ samples. ∎

## 4.1 Proof of Lemma 4.1

Let $\mathcal{N}(\mu, \Sigma)$ denote the target distribution, which we are to encode.

**Remark 4.3** *The case of rank-deficient $\Sigma$ can easily be reduced to the case of full-rank $\Sigma$. If the rank of $\Sigma$ is $r < d$, then any $X \sim \mathcal{N}(\mu, \Sigma)$ lies in some affine subspace $\mathcal{S}$ of dimension $r$. With high probability, the first $d$ samples from $\mathcal{N}(\mu, \Sigma)$ uniquely identify $\mathcal{S}$. We encode $\mathcal{S}$ using these samples, and for the rest of the process we work in this affine subspace. Hence, we may assume $\Sigma$ has full rank $d$.*

To prove Lemma 4.1, we will need the following result from the random matrix theory literature [cf. 16, Corollary 4.1]. Let $S^{d-1} := \{ y \in \mathbb{R}^d : \|y\| = 1 \}$ and $B_2^d := \{ y \in \mathbb{R}^d : \|y\| \leq 1 \}$. We use the notation $\frac{1}{20} B_2^d$ to denote the set of $d$-dimensional vectors with Euclidean norm at most $1/20$. The convex hull of a set $T$ is denoted by $\text{conv}(T)$.

**Lemma 4.4** *Let $q_1, \ldots, q_m$ be i.i.d. samples from $\mathcal{N}(0, I_d)$, and let $T := \{ \pm q_i : \|q_i\| \leq 4\sqrt{d} \}$. Then for a large enough constant $C > 0$, if $m \geq Cd(1 + \log d)$ then*

$$\mathbf{Pr}\left[ \frac{1}{20} B_2^d \not\subseteq \text{conv}(T) \right] \leq 1/6.$$

Note that the lemma can be improved to require only $m \geq Cd$ samples [see 16, Corollary 4.1], but this would not improve our final bound.

The remainder of the proof amounts to showing that with only a small number of additional bits, we can approximate the mean and each eigenvector of the covariance matrix as a linear combination of a subset of the drawn samples.

Suppose $\Sigma = \sum_{i=1}^d v_i v_i^\mathsf{T}$, where the $v_i$ vectors are orthogonal. Let $\Psi := \sum_{i=1}^d v_i v_i^\mathsf{T}/\|v_i\|$. Note that both $\Sigma$ and $\Psi$ are positive definite, and that $\Sigma = \Psi^2$. Moreover, it is easy to see that $\Sigma^{-1} = \sum_{i=1}^d v_i v_i^\mathsf{T}/\|v_i\|^4$ and $\Psi^{-1} = \sum_{i=1}^d v_i v_i^\mathsf{T}/\|v_i\|^3$.

**Lemma 4.5** *Let $C > 0$ be a sufficiently large constant. Given $m = 2Cd(1 + \log d)$ samples $S$ from $\mathcal{N}(\mu, \Sigma)$, with probability at least $2/3$, one can encode vectors $\widehat{v}_1, \ldots, \widehat{v}_d, \widehat{\mu} \in \mathbb{R}^d$ satisfying*

$$\|\Psi^{-1}(\widehat{v}_j - v_j)\| \leq \varepsilon/24d^2 \qquad \forall j \in [d],$$

*and $\|\Psi^{-1}(\widehat{\mu} - \mu)\| \leq \varepsilon/2$, using $O(d^2 \log(2d) \log(d/\varepsilon))$ bits and the points in $S$.*

Lemma 4.1 now follows immediately from the following lemma

**Lemma 4.6** *Suppose $\Sigma = \Psi^2 = \sum_{i \in [d]} v_i v_i^\mathsf{T}$, where the $v_i$ are orthogonal and $\Sigma$ is full rank, and that $\|\Psi^{-1}(\widehat{\mu} - \mu)\| \leq \zeta$, and that $\|\Psi^{-1}(\widehat{v}_j - v_j)\| \leq \rho \leq 1$ holds for all $j \in [d]$. Then,*

$$\text{TV}\left( \mathcal{N}\left( \mu, \sum_{i \in [d]} v_i v_i^\mathsf{T} \right), \mathcal{N}\left( \widehat{\mu}, \sum_{i \in [d]} \widehat{v}_i \widehat{v}_i^\mathsf{T} \right) \right) \leq \sqrt{9d^3\rho^2 + \zeta^2}/2.$$

# 5 Discussion

A central open problem in distribution learning and density estimation is characterizing the sample complexity of learning a distribution class. An insight from supervised learning theory is that

the sample complexity of learning a class (of concepts, functions, or distributions) is typically proportional to (some notion of) intrinsic dimension of the class divided by $\varepsilon^2$, where $\varepsilon$ is the error tolerance. For the case of agnostic binary classification, the intrinsic dimension is captured by the VC-dimension of the concept class (see [21, 4]). For the case of distribution learning with respect to 'natural' parametric classes, we expect this dimension to be equal to the number of parameters. This is indeed true for the class of Gaussians (which have $d^2$ parameters) and axis-aligned Gaussians (which have $d$ parameters), and we showed in this paper that it holds for their mixtures as well (which have $kd^2$ and $kd$ parameters, respectively).

In binary classification, the combinatorial notion of Littlestone-Warmuth compression has been shown to be sufficient [15] and necessary [18] for learning. In this work, we showed that the new but related notion of distribution compression is sufficient for distribution learning. Whether the existence of compression schemes is necessary for learning an arbitrary class of distributions remains an intriguing open problem.

It is worth mentioning that while it may first seem that the VC-dimension of the Yatracos set associated with a class of distributions can characterize its sample complexity, it is not hard to come up with examples where this VC-dimension is infinite while the class can be learned with finite samples. Covering numbers do not characterize the sample complexity either: for instance the class of Gaussians does not have a finite covering number in the TV metric, nevertheless it is learnable with finitely many samples.

A concept related to compression is that of *core-sets*. In a sense, core-sets can be viewed as a special case of compression, where the decoder is required to be the empirical error minimizer. See [17] for using core-sets in maximum likelihood estimation.

### Acknowledgments

We thank Yaoliang Yu for pointing out a mistake in an earlier version of this paper, and Luc Devroye for fruitful discussions. Abbas Mehrabian was supported by a CRM-ISM postdoctoral fellowship and an IVADO-Apogée-CFREF postdoctoral fellowship. Nicholas Harvey was supported by an NSERC Discovery Grant. Christopher Liaw was supported by an NSERC graduate award. Yaniv Plan was supported by NSERC grant 22R23068.

### Addendum

The lower bound of Theorem 1.2 was recently improved in a subsequent work [8] from $\Omega(kd^2/\varepsilon^2 \log^3(1/\varepsilon))$ to $\Omega(kd^2/\varepsilon^2 \log(1/\varepsilon))$ using a different construction.

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
