[Reviews · NeurIPS 2018]

Reviewer 1



This paper provides near-optimal bounds on the sample complexity of estimating Gaussian mixture models under the total variation distance. A novel scheme of leveraging results on compressibility to give bound on learnability is employed, allowing for an improvement of O(d) vs. existing bounds. The results are well motivated and presented, and provide an improved understanding of the complexity of learning GMMs, a fundamental classical task in ML. Some suggestions: In terms of writing, the paper seems to peter out towards the end, with little text connecting a list of lemmas and remarks. Adding additional connective tissue and intuitive explanation here, possibly including a conclusion/discussion section, would improve readability. Several of the lemmas may be moved to the supplement to achieve this. Edit: The authors in their response addressed my concerns, so I reiterate my recommendation.

Reviewer 2



This is a very well-written paper on density estimation, measured in the L1 - or total variation - sense. It addresses a surprising gap in the litterature, and shows the subtle differences between density learning for this problem and other versions such as nonparametric density learning for smooth densities, or learning the parameters of a mixture of Gaussians. The approach is, to my knowledge, original and correct. This is significant for the theoretical understanding of this problem, and this is a very sophisticated approach to do so.

Reviewer 3



This paper is well written and issues a classical and well known studied problem from a new point of view. The compression based analysis which is indeed deep, although less popular, is gaining more attention on the last couple of years. This paper join this line of work by extending it, proving some interesting properties if sample-compression schemes (such as the them being close under mixture or product) and go on and demonstrates the power of the obtained results by proving state of the art sample complexity upper bound for mixture of Gaussians. I think that the NIPS community will indeed benefit from this paper. Remarks: - there exists a line of work regarding the generalization guarantees of compression based learners. Most of them to the classification case but some for more general setting. How does your results differ from those? See for example On statistical learning via the lens of compression, PAC-Bayesian Compression Bounds on the Prediction Error of Learning Algorithms for Classification, Nearly optimal classification for semimetrics, Sample Compression for Real-Valued Learners, Adaptive Learning with Robust Generalization Guarantees, - Although the “robust compression implies agnostic learning” is indeed of interest it isn’t in use on the specific case of mixture of Gaussians, I think that it should be stated more clearly as I got a bit of confused, especially as the proof is given for that case and not for the (more simple) realizable case - On line 317 you use a notion of a d dimensional ball. I know that this is standard notion but still it may be of help to mention you intention to make it more clear what each parameter means (so for example when you multiply the ball, you mean just scaling it?